# Cholesterol Homeostasis Modulates Platinum Sensitivity in Human Ovarian Cancer

**DOI:** 10.3390/cells9040828

**Published:** 2020-03-30

**Authors:** Daniela Criscuolo, Rosario Avolio, Giovanni Calice, Chiara Laezza, Simona Paladino, Giovanna Navarra, Francesca Maddalena, Fabiana Crispo, Cristina Pagano, Maurizio Bifulco, Matteo Landriscina, Danilo Swann Matassa, Franca Esposito

**Affiliations:** 1Department of Molecular Medicine and Medical Biotechnology, University of Naples Federico II, 80131 Naples, Italy; daniela.criscuolo@unina.it (D.C.); simona.paladino@unina.it (S.P.); vanna.navarra@libero.it (G.N.); pagano.cris@gmail.com (C.P.); maurizio.bifulco@unina.it (M.B.); 2CRG - Centre for Genomic Regulation, 08003 Barcelona, Spain; rosario.avolio@crg.eu; 3Laboratory of Pre-Clinical and Translational Research, IRCCS, Referral Cancer Center of Basilicata, 85028 Rionero in Vulture, Italy; giovanni.calice@crob.it (G.C.); francesca.maddalena@crob.it (F.M.); fabiana.crispo@crob.it (F.C.); matteo.landriscina@unifg.it (M.L.); 4Institute Experimental Endocrinology and Oncology “Gaetano Salvatore”, National Research Council (IEOS-CNR), 80131 Naples, Italy; chilaez@hotmail.com; 5Medical Oncology Unit, Department of Medical and Surgical Sciences, University of Foggia, 7100 Foggia, Italy

**Keywords:** ovarian cancer, TRAP1, cholesterol homeostasis, platinum resistance

## Abstract

Despite initial chemotherapy response, ovarian cancer is the deadliest gynecologic cancer, due to frequent relapse and onset of drug resistance. To date, there is no affordable diagnostic/prognostic biomarker for early detection of the disease. However, it has been recently shown that high grade serous ovarian cancers show peculiar oxidative metabolism, which is in turn responsible for inflammatory response and drug resistance. The molecular chaperone TRAP1 plays pivotal roles in such metabolic adaptations, due to the involvement in the regulation of mitochondrial respiration. Here, we show that platinum-resistant ovarian cancer cells also show reduced cholesterol biosynthesis, and mostly rely on the uptake of exogenous cholesterol for their needs. Expression of FDPS and OSC, enzymes involved in cholesterol synthesis, are decreased both in drug-resistant cells and upon TRAP1 silencing, whereas the expression of LDL receptor, the main mediator of extracellular cholesterol uptake, is increased. Strikingly, treatment with statins to inhibit cholesterol synthesis reduces cisplatin-induced apoptosis, whereas silencing of LIPG, an enzyme involved in lipid metabolism, or withdrawal of lipids from the culture medium, increases sensitivity to the drug. These results suggest caveats for the use of statins in ovarian cancer patients and highlights the importance of lipid metabolism in ovarian cancer treatment.

## 1. Introduction

Ovarian cancer (OC) encompasses a heterogenous group of malignancies differentiated by cell/site of origin, pathological grade, risk factors, prognosis, and treatment. Among all OC subtypes, non-epithelial neoplasias are the least aggressive, whereas epithelial malignancies are the most common ones, accounting for 90% of all cases [1]. Epithelial cancers (EOC) are classified by tumor cell histology as serous (52%), endometrioid (10%), mucinous (6%), or clear cell (6%), with the rest being rarer subtypes or unspecified [1]. It has been proposed to further group EOC as type I or type II based on clinicopathologic factors, with type I classified as low-grade malignancies, except clear cell carcinomas, and accounting for only a small fraction of deaths, and type II considered high grade tumors and characterized by aggressiveness and low survival [2]. The predominant high-grade serous carcinomas are mostly diagnosed at stage III (51%) or IV (29%), reflecting their poorly symptomatic and aggressive nature. Consequently, the five-year cause-specific survival for serous carcinoma is 43%, compared with 82%, 71%, and 66% for endometrioid, mucinous, and clear cell carcinoma, respectively [1]. Surprisingly, large-scale genomic analysis of high grade serous ovarian cancers (HGSOC) has uncovered only a few recurrently mutated genes, such as TP53 and BRCA1/BRCA2 [3]. Given the high heterogeneity and the complexity progressively unveiled in recent years, and despite novel therapeutic approaches, only poly(ADP-ribose) polymerase (PARP) inhibitors show significant clinical activity in women with BRCA1 or BRCA2 mutations, whereas most targeted therapies have not shown substantial results [4]. Therefore, although advances are being made, platinum-based therapy combined with taxanes is still the primary choice for the treatment of ovarian cancers, which remains the most fatal gynecologic malignancy [5]. Indeed, while approximately 75% of patients initially respond to the platinum/paclitaxel-based chemotherapy, most of them relapse and progressively develop platinum resistance. Interestingly, it has been shown that a peculiar metabolic remodeling is associated to acquisition of chemoresistance in HGSOC. Indeed, advanced-stage HGSOC cells surprisingly remodel their metabolism toward a more oxidative one, as opposed to the canonical “Warburg” metabolism, characterized by prevalent glycolysis even under aerobic conditions [6]; furthermore, a more oxidative OC cell phenotype correlates with drug resistance [7]. Mechanistically, we previously reported that the metabolic switch triggers an inflammatory response, with endogenous production of proinflammatory cytokines, which ultimately leads to the upregulation of drug-resistance factors of the multidrug resistance (MDR) family, through an unknown mechanism [7]. A pivotal player in this scenario is the molecular chaperone Tumor necrosis factor Receptor Associated Protein 1 (TRAP1), a heat shock protein (HSP) 90 family member that binds respiratory complexes into the mitochondria. Upon this interaction, oxidative phosphorylation is reduced, when glucose is available [8]. Indeed, in recent years, TRAP1 has proved to be an interesting biomarker in OC since its expression inversely correlates with stage and grade in HGSOC, and directly correlates with an increased survival [7]. TRAP1 levels are also reduced in metastatic sites compared to the primary tumor, and a downregulation of its expression correlates with epithelial-mesenchymal transition, which in turn indicates chemoresistance [9]. To date, the mechanisms of the whole metabolic remodeling observed in drug resistant OC cells still needs to be understood, and the mechanisms ultimately responsible for OC drug resistance are partially unknown. Herein, we show a novel metabolic alteration in cisplatin-resistant HGSOC cells, i.e., the dysregulation of the cholesterol metabolism. This is due to inhibition of cholesterol biosynthesis, with cells mostly relying on its uptake from the extracellular environment. Remarkably, we show that TRAP1 downregulation in drug-sensitive cells mimic some of the effects observed in the resistant counterpart. Finally, an inhibition of the cholesterol biosynthetic pathway by statins induces platinum resistance, whereas silencing of enzymes involved in lipid uptake or withdrawal of extracellular lipids increase drug sensitivity.

## 2. Materials and Methods

### 2.1. Cell Cultures

The paired HGSOC cell lines PEA1/PEA2 and PEO14/PEO23 have been described elsewhere [10]. TRAP1-stable interfered cells were obtained by transfecting PEA1 cells with TRAP1 (TGCTGTTGACAGTGAGCGACCCGGTCCCTGTACTCAGAAATAGTGAAGCCACAGATGTATTTCTGAGTACAGGGACCGGGCTGCCTACTGCCTCGGA) or scrambled (sequence containing no homology to known mammalian genes) short hairpin RNAs (shRNA) (Open Biosystem) and isolating positive clones by selection with Puromycin as previously described [7]. All lines were maintained in RPMI 1640 media with 10% fetal bovine serum, glutamine and Normocin (Invivogen), at 37 °C, 5% CO_2_.

### 2.2. Transfection Procedures

Transient silencing was performed with siRNAs targeting: TRAP1 (Qiagen S.r.l., Milano, Italy; cat. no. SI00115150, target sequence CCCGGTCCCTGTACTCAGAAA), LIPG (Qiagen S.r.l., Milano, Italy; cat. no. SI03039551, target sequence AACGATGTCTTGGGATCAATT) and nontargeting control siRNA (Qiagen S.r.l., Milano, Italy; cat. no. SI03650318). Transfections were performed using HiPerFect Transfection Reagent (Qiagen S.r.l., Milano, Italy), according to manufacturer’s protocol.

### 2.3. RNA Extraction and Real-Time Reverse Transcriptase-Polymerase Chain Reaction (RT-PCR)

Total RNA extraction procedures were performed by using TRI Reagent (Merck Life Science S.r.l., Milano, Italy; product code T9424), following the manufacturer’s instruction. For first-strand synthesis of cDNA, 1 μg of RNA was used in a 20-μL reaction mixture by using a SensiFast cDNA synthesis kit (Bioline, London, UK). For real-time PCR analysis, 0.4 μL of cDNA sample was amplified by using the SensiFast Syber (Bioline, London, UK) in an iCycler iQ Real-Time Detection System (Bio-Rad Laboratories GmbH, Segrate, Italy). The reaction conditions were 95 °C for 2 min followed by 40 cycles of 5 s at 95 °C and 30 s at 60 °C. Actin was chosen as the internal control. In PCR analyses performed upon TRAP1 silencing, RNAs were collected 72 h after siRNA transfection. The following primers (Table 1) were used for PCR analysis.

### 2.4. Gene Expression Analyses

Total RNAs from PEA1 cells were extracted using TRIzol reagent (Thermo Fisher Scientific, Monza, Italy) after 72 h from transfection with TRAP1-directed or control siRNA. RNA concentration was evaluated with a NanoDrop 2000c spectrophotometer (Thermo Fisher Scientific, Monza, Italy); its quality was assessed with a Bioanalyzer 2100 (Agilent Technologies, Santa Clara, CA, USA). For each sample, 300 ng of total RNA were reverse transcribed and used for synthesis of cDNA and biotinylated cRNA according to the Illumina TotalPrep RNA amplification kit protocol (Thermo Fisher Scientific, Monza, Italy). Of each cRNA, 750 ng was hybridized on an Illumina HumanHT12 v4.0 Expression BeadChip array (Illumina, San Diego, CA, USA); staining was performed according to standard protocol supplied by Illumina. BeadChips were dried and scanned with an Illumina HiScanSQ system (Illumina, San Diego, CA, USA). Analysis was performed in three replicates for control samples and two replicates for TRAP1-silenced samples. Raw signal intensity data from Illumina HumanHT-12_V4_0_R2 microarrays were normalized using the neqc function in the limma package, and low-quality annotation probes were excluded. Differentially expressed genes (DEGs) were obtained by a moderated *t*-test on the linear model fit of the microarray data. DEGs with *p*-values <0.05 were retained. All the steps performed according to the “microarray analysis” best practice using well-known packages in R (https://www.r-project.org) [11]. Microarray data are publicly available at the Gene Expression Omnibus (GEO) database under accession number GSE144248.

### 2.5. Gene Set Enrichment Analyses

The enrichment analysis was executed by the GSEA (Gene Set Enrichment Analysis) function in the clusterProfiler package [12] and on the base of the Hallmark gene sets collection in MsigDB (Molecular Signatures Database) [13]. The results are shown in Figure 1 on significant categories (adjusted *p*-value <0.05) and on absolute enrichment score (ES) > 0.4.

### 2.6. Western Blot Analyses

Equal amounts of protein from cell lysates and tumor specimens were subjected to SDS-PAGE (Sodium Dodecyl Sulphate - PolyAcrylamide Gel Electrophoresis) and transferred to a Polyvinylidene fluoride (PVDF) membrane (Merck Life Science S.r.l., Milano, Italy). The following antibodies were used: anti-TRAP1 (Santa Cruz Biotechnology, Inc., Dallas, TX, USA; sc-13557), anti-β-Actin (Santa Cruz Biotechnology, Inc., Dallas, TX, USA; sc-69879), anti-LDLR (Santa Cruz Biotechnology, Inc., Dallas, TX, USA; sc-18823), anti-FDPS (OriGene, Herford, Germany; AP12197PU-N), and anti-OSC (Santa Cruz Biotechnology, Inc., Dallas, TX, USA; sc-514507). Quantitative estimation was performed by measuring densitometric band intensity using ImageJ [14].

### 2.7. Incorporation of ^14^C Acetate

Actively growing, subconfluent PEA1 and PEA2 cells and siRNA-transfected PEA1 cells (see “transfection procedures” section for details) were incubated with 10 μL of 2-^14^C-acetate([14C]- acetate) (Specific activity: 59 mCi/mmol, Perkin Elmer, Waltham, MA, USA) for 12 h. Cells were washed and harvested with cold Phosphate-buffered saline (PBS), resuspended in 500 μL of isopropanol and sonicated. After centrifugation, supernatants were dried and dissolved in 50 μL of chloroform, whereas pellets were solubilized in 500 μL of 0.1 M NaOH for the protein assay. In all, 10 μL of each sample were loaded on Thin Layer Chromatography—Silica gel-60G (Millipore, Billerica, MA, USA) in presence of 1,2-3H(N)cholesterol standard (Specific activity 60.0 Ci/mmol, NEN products—DuPont, Boston, MA, USA) and developed using hexane/diethylether/acetic acid (70:30:1) as a solvent system. Membranes were exposed to X-ray film and film developed, spots were quantified by densitometry using a digital imaging system (Image Lab software, Berkeley, CA, USA).

### 2.8. Microscopy

Twenty-four hours after seeding on coverslips, cells were fixed with 4% (*w*/*v*) paraformaldehyde in PBS for 15 min, then stained with filipin III for 1 h at room temperature. After Filipin staining, immunofluorescence assays were performed as previously described [15], cells were permeabilized in blocking solution containing Triton (Fetal Bovine Serum 5%, Bovine Serum Albumin 0.4%, Triton 0.1% in PBS), then hybridized with primary antibodies for 2 h, washed three times in PBS, and hybridized with secondary antibodies for 1 h. The following primary and secondary antibodies were used: anti-GM130 (Cell Signaling Technology, 12480); anti-CAV1 (Santa Cruz Biotechnology, Inc., Dallas, TX, USA; sc-894); AlexaFluor 488-conjugated anti-Rabbit (Jackson ImmunoResearch Europe Ltd., Cambridge, UK; 111-545-003); Dylight594-conjugated anti-Mouse (Abcam, Cambridge, UK; ab96873). For labeling lysosomes, cells were incubated with a Lysotracker probe (Thermo Fisher Scientific, Monza, Italy) in complete medium for 1 h at 37 °C before fixation. Images were collected using the Leica Thunder Imaging System (Leica Microsystems Srl, Buccinasco, Italy) equipped with Leica DFC9000GTC camera and a planapo 63× oil immersion (NA 1.4) objective lens. A fluorescence LED light source and appropriate excitation and emission filters were used. Images were acquired taking Z-slices from the top to the bottom of the cell by using the same setting (LED source power, exposure time) and the Small Volume Computational Clearing (SVCC) mode for the different cell lines and in all experimental conditions (control and TRAP1-interfered cells). Quantification analyses were carried out using Leica Thunder software. The mean fluorescence intensities were measured by drawing regions of interest (ROIs) of same size and corrected for the background as previously described [15].

### 2.9. Cell Treatments and Apoptosis and Viability Assays

In cell viability and apoptosis assays, PEA1 cells were treated for 24 h with Lovastatin (5 μM), and then contemporary with Lovastatin (5 μM) and cisplatin (20 μM) for additional 48 h. Apoptosis was measured using the Caspase-Glo 3/7 assay (Promega Italia s.r.l, Milano, Italy; product code G8090) and was performed according to the manufacturer’s instructions. In lipid deprivation experiments, 24 h after seeding, the culture medium was replaced with RPMI 1640 supplemented with charcoal-stripped serum (Thermo Fisher Scientific, Monza, Italy; product code A3382101) for a subsequent 48 or 96 h. Cell viability was measured by MTT (3-(4,5-dimethylthiazol-2-yl)-2,5-diphenyltetrazolium bromide) assay by using the in vitro toxicology assay kit (Merck Life Science S.r.l., Milano, Italy; product code TOX1-1KT), following the manufacturer’s instructions. For oxidative stress induction, PEA1 cells were treated with hydrogen peroxide (20 µM) for 9 h.

### 2.10. Statistics

The two-tailed unpaired Student’s *t*-test was used to establish the statistical significance of caspase 3/7 activation in apoptosis assays and of changes in gene expression levels compared to controls in real-time PCR experiments, and changes in cell viability in MTT assays and of densitometric band intensity in western blots.

## 3. Results

### 3.1. TRAP1 Expression and Chemoresistance Correlate with Metabolism, Inflammation, and Lipid Homeostasis

We have previously demonstrated that the expression of the molecular chaperone TRAP1 in HGSOC inversely correlates with stage, grade, and progression to metastatic disease and directly correlates with survival [7,9]. Indeed, TRAP1 silencing in HGSOC cells induces chemoresistance through an oxidative metabolism-induced inflammatory response, but the molecular mechanisms underlying this phenomenon are yet to be elucidated. To this aim, we performed a whole genome gene expression analysis in the cisplatin-sensitive HGSOC cell line PEA1 upon siRNA-mediated TRAP1 interference (Appendix A). We identified 1001 differentially expressed genes with a *p*-value < 0.01, 571 (57%) of which were upregulated in TRAP1-silenced cells and 430 (43%) downregulated (Figure 1A). TRAP1 itself resulted downregulated with a fold change of 0.23, proving the efficacy of the transfection procedure. Notably, the most upregulated gene was Colony Stimulating Factor 2 (CSF2) (fold change = 3.22), which is among the cytokines we have previously identified as induced by metabolic shift towards oxidative phosphorylation induced in platinum-sensitive PEA1 cells by glucose deprivation, treatment with the mitochondrial uncoupler FCCP (Carbonyl cyanide-4-(trifluoromethoxy)phenylhydrazone), and TRAP1 silencing itself [7]. Similarly, Interleukine 6 (IL6) and Plasminogen Activator Urokinase (PLAU) upregulation were also confirmed (Figure 1A). Gene set enrichment analyses performed on this dataset confirmed the causal nexus between TRAP1 silencing and hallmarks as interleukin signaling and inflammatory response (Figure 1B). Among the regulated hallmarks, cholesterol homeostasis was also significantly enriched. Accordingly, enrichment analyses performed by Enrichr [16] showed that the most enriched pathway, according to the Wikipathways database [17], was correlated to cholesterol biosynthesis (Figure 1C), with “cholesterol metabolism” also enriched. Starting from this evidence, we focused our attention on the differentially regulated genes that are involved in cholesterol synthesis and metabolism: 3-Hydroxy-3-Methylglutaryl-CoA Synthase 1 (HMGCS1), Lipase G (LIPG), Niemann-Pick C1 Protein (NPC1), Methylsterol Monooxygenase 1 (MSMO1), and Sterol-C5-Desaturase (SC5D). We investigated the regulation of this shortlist of genes by comparing their expression in control OC cells with cells transfected with TRAP1-directed siRNA in two different cisplatin-sensitive cell lines, PEA1 and PEO14, and by comparing their expression levels in the platinum-resistant PEA2 cells with the matched-sensitive counterpart PEA1 (Figure 1D). Of note, all the target except SC5D modulated their expression concordantly in all the pairings, and according to microarray indications. In particular, LIPG was significantly upregulated in PEO14 cells upon TRAP1 silencing, whereas MSMO1 and NPC1 were, respectively, significantly downregulated and upregulated in the cisplatin-resistant PEA2 cells compared to the matched-sensitive PEA1. The protein encoded by the LIPG gene has phospholipase activity and in endothelial cells it is involved in the maintenance of cholesterol homeostasis [18], MSMO1 is a methyl-sterol oxidase that is believed to function in cholesterol biosynthesis [19], NPC1 encodes for a protein involved in intracellular cholesterol trafficking [20], and SC5D catalyzes the conversion of lathosterol into 7-dehydrocholesterol in the last steps of cholesterol biosynthesis. Taken together, all these data highlight the importance of lipid metabolism (in particular of cholesterol) in the transition from a chemosensitive to a drug-resistant disease.

Our findings confirm previous studies on transcriptional analysis of a matched cell line series from three patients with HGSOC before and after development of clinical platinum resistance [21], showing that cholesterol biosynthesis is one of the most dysregulated pathways in the drug-resistant cells [7]. We therefore decided to explore the cholesterol pathway by analyzing the expression levels of some crucial component of its synthesis, either in the chemosensitive PEA1 and PEO14 cells following siRNA-mediated TRAP1 silencing, and in PEA2 cells compared to PEA1 (Figure 2A). Results showed no significant regulation of these additional genes involved in cholesterol biosynthesis upon TRAP1 silencing. On the other hand, Low Density Lipoprotein (LDL) Receptor (LDLR), which encodes for the protein responsible for the receptor-mediated endocytosis of LDL, showed a dramatic upregulation in PEA2 cells, whereas enzymes involved in cholesterol biosynthesis such as mevalonate kinase (MVK), oxidosqualene cyclase (OSC), and 24-dehydrocholesterol reductase (DHCR24) were significantly decreased. Remarkably, and in support of our data, DHCR24, which catalyzes the reduction of the delta-24 double bond of sterol intermediates during cholesterol biosynthesis, was already reduced in the previous microarray analysis [21]. This is particularly relevant for DHCR24, a terminal enzyme for the cholesterol biosynthesis, shared by both the Bloch [22] and Kandutsch-Russell [23] pathways. Western blot analyses confirmed the upregulation of the LDLR protein and the downregulation of OSC in PEA2 cells compared to PEA1, but also showed that, although the mRNA expression is unchanged, the Farnesyl Diphosphate Synthase (FDPS) protein is reduced (Figure 2B), as an additional and independent regulatory mechanism for pathway downregulation. Similar results were obtained by comparing PEO14 cells with their resistant counterpart PEO23, with significant upregulation of LDLR and slight reduction of FDPS and OSC. TRAP1 silencing in PEA1 at least partially mimicked the regulation observed in the transition to chemoresistance, with reduced expression of FDPS and OSC (Figure 2B). Altogether, these results suggest that a rewiring of the cholesterol pathway may be an important step in the transition to a drug-resistant state in OC cells, and that TRAP1 may contribute.

Based on these data, we decided to investigate the cholesterol biosynthetic pathway in drug-sensitive and drug-resistant HGSOC cells and the impact of TRAP1 expression on its regulation. To this aim, we measured incorporation of ^14^C acetate into cholesterol by thin-layer chromatography [24]. Results showed that the cholesterol synthesis is dramatically reduced in PEA2 cells compared to PEA1, and that TRAP1 silencing in the latter also decreases the incorporation of acetate (Figure 2C). These data are consistent with the reduced expression levels of the enzymes involved in cholesterol synthesis.

### 3.2. Cholesterol Distribution Is Altered in Chemoresistant Cells

Considering that LDLR is upregulated in chemoresistant cells, whereas the biosynthetic pathway is inhibited, we decided to dissect cholesterol distribution in cisplatin-sensitive and cisplatin-resistant cells by microscopy analyses. To this aim, cells were labeled with filipin III, a polyene macrolide widely used to localize and quantitate unesterified cholesterol (Figure 3). As expected for mammalian cells [25], cholesterol is enriched in the plasma membrane of the four ovary cell lines (Figure 3). Interestingly, higher levels of cholesterol on the plasma membrane of chemoresistant PEA2 and PEO23 compared to their sensitive counterpart PEA1 and PEO14 were observed (Figure 3A). The accumulation of cholesterol on the cell membrane of cisplatin-resistant cells is also supported by colocalization with caveolin -1 (CAV-1), a cholesterol binding protein that is the main component of caveolae in plasma membranes [25] (Figure 3B). Similar results were observable in stable TRAP1 knockdown PEA1 clones compared to nontargeting shRNA controls (Appendix A).

Furthermore, we investigated the intracellular distribution of cholesterol by double immunofluorescence assays using different markers of subcellular compartments. It is worth noting that a fraction of intracellular cholesterol is enriched in the Golgi membranes labeled with GM130, a Golgi marker, both in cisplatin-sensitive and -resistant cells (Figure 4A). In addition, in both cell lines Golgi filipin-positive structures display comparable distribution in the perinuclear region and similar fluorescence intensity, indicating no alteration of the trafficking of newly synthetized cholesterol. Furthermore, intracellular filipin-positive dots have also been found to colocalize with lysotracker dye, a reagent that has affinity for the low pH of lysosomal compartments, both in PEA1 and PEA2 cells (Figure 4B). A lot of these structures were observed in the chemoresistant cells, supporting that much cholesterol is taken up by these cells and traffic toward lysosomes, consistently with higher expression of LDLR.

### 3.3. Reduced Cholesterol Biosynthesis and Increased Cholesterol Uptake from Extracellular Environment Induce Chemoresistance

Based on the observation that chemoresistant cells downregulate endogenous pathways of cholesterol synthesis, which mostly rely on extracellular cholesterol uptake for their needs, we wondered if a modulation of the biosynthetic pathways has an impact on drug resistance. To investigate this issue, cells were treated with Lovastatin (5 μM), an inhibitor of cholesterol synthesis, for 24 h, before adding cisplatin (20 μM) for an additional 48 h, and measured apoptosis by caspase 3/7 activity through a luminescent assay. Results showed that pre-treatment with Lovastatin significantly reduced cisplatin-induced apoptosis in both the drug-sensitive PEA1 and PEO14 cells, with the first cell line being more dramatically affected, whereas drug-resistant PEA2 cells showed no significant caspase activation at the same cisplatin concentration (Figure 5A). Indications obtained by cytofluorimetric analyses of AnnexinV/Propidium Iodide staining and immunoblot analyses of PARP-1 cleavage yielded similar results (Appendix A).

Our results suggest that chemoresistant cells reduce cholesterol synthesis in order to earn drug resistance, and, in parallel, activate compensatory uptake of exogenous cholesterol for membrane composition. Future studies will address mechanisms involved in this process. Starting from this, we measured viability of chemosensitive and chemoresistant cells upon withdrawal of lipids from extracellular medium, by culturing cells in medium supplemented with lipid-deprived serum for 48 and 96 h. Interestingly, we found that the chemoresistant PEA2 cells, showing reduced biosynthesis and increased levels of LDLR, are not sensitive to lipid deprivation, but rather slightly increase their viability (Figure 5B). Accordingly, western blot analysis demonstrate that those conditions trigger significant re-expression of OSC and a slight increase in FDPS levels, suggesting a feedback reactivation of the biosynthetic pathway (Figure 5C). We therefore measured cisplatin-induced apoptosis in PEA2 cells following lipid withdrawal from serum supplementing the culture medium. Strikingly, lipid deprivation significantly increased caspase 3/7 activation following cisplatin treatment (40 μM) (Figure 5D), further supporting the hypothesis that repression of the biosynthetic pathway is functional to the acquisition of drug resistance, and that its re-activation can restore sensitivity. Significantly, an analysis of The Cancer Genome Atlas (TCGA) database, performed with GEPIA (Gene Expression Profiling Interactive Analysis) [26], revealed that low expression of the biosynthetic enzyme FDPS correlates with reduced overall survival in OC patients (Figure 5E), whereas, on the contrary, high expression of LDLR is associated with worse survival (Figure 5F), which is in line with our in vitro observations.

### 3.4. Oxidative Stress Induces Inflammatory Response and Remodeling of Lipid Metabolism

The data shown above, along with our previous observations [7] strongly suggest that OC cells remodel their metabolism reducing lipid biosynthetic pathways and increasing catabolic pathways through the tricarboxylic acid cycle and the oxidative phosphorylation, gaining survival advantage in terms of resistance to apoptosis induced by anticancer drugs. Our analyses highlighted the potential role of LIPG in such metabolic remodeling. For this reason, we explored the impact of LIPG expression on PEA1 sensitivity to cisplatin treatment. Remarkably, siRNA-mediated LIPG silencing significantly increased caspase 3/7 activity upon cisplatin treatment (20 μM for 48 h) (Figure 6A). Interestingly, it has been shown that the LIPG gene is transcriptionally activated by NF-kappaB, a pivotal mediator of inflammatory responses [27]. Considering that we have previously shown that HGSOC cells acquire cisplatin resistance through oxidative metabolism-induced inflammation, we wondered whether the induction of oxidative stress in these cells is sufficient to induce the expression of any of the proinflammatory cytokines previously identified (CSF2, IL6, IL8), the ABC transporter potentially responsible for drug resistance (TAP1) [7], and LIPG itself. To this aim, we measured the expression of these genes in PEA1 cells by qPCR following treatment with H_2_O_2_ (20 μM) for 9 h. The results show that all of the genes mentioned above are, to various extents, induced upon treatment, but only CSF2 and IL8 were significantly upregulated (Figure 6B). Further investigations are needed to clarify the possible involvement of LIPG in the inflammatory response induced by oxidative stress.

## 4. Discussion

EOC is usually diagnosed at an advanced stage, as early stages have no obvious symptoms, and to date, the efficacy of screening has not been demonstrated in prospective randomized controlled trials. Among all the subtypes, HGSOC is responsible for the majority of deaths and, even though the introduction of PARP inhibitors led to large advances, improvements are still needed for a more successful therapy. At present, the first-line therapy is mainly based on combinations of platinum derivatives with taxanes, quite effective initially, but which frequently end up with disease relapse and acquisition of platinum resistance. We have previously demonstrated that drug-resistant OC cells, contrarily to most cancer types, show an unexpected activation of oxidative metabolism. This metabolic environment contributes to chemoresistance by eliciting an inflammatory response that, in turn, stimulates the expression of drug-resistance factors [7]. A pivotal contributor of such metabolic adaptations is the molecular chaperone TRAP1, which binds and regulates the activity of respiratory complexes, thus exerting oncogene/oncosuppressor functions depending on the metabolic phenotype of each tumor type [28]. Metabolic reprogramming is an emerging concept in tumor biology, now recognized as one of the hallmarks of cancer, and accumulating evidence suggests the possibility to target pathways of energy metabolism as a therapeutic strategy to overcome drug resistance [29]. Herein, we further deepened the analysis of the metabolic rewiring in chemoresistant OC cells by taking advantage of a model system based on patient-derived couples of cell lines stabilized before and after clinically-acquired drug resistance upon platinum-based treatments. Whole genome gene expression analyses of these cells and of TRAP1 knock-down cells highlighted the dysregulation of the cholesterol biosynthetic pathway. Accordingly, a small subset of genes, whose expression is altered in OC cells following TRAP1 silencing, was enriched in genes involved in lipid homeostasis. Cholesterol homeostasis is required for the normal growth of eukaryotic cells, given the important role of cholesterol within cell membranes, where it regulates membrane fluidity, signaling initiation, and cell adhesion to the extracellular matrix [30]. Studies have shown that cholesterol synthesis is increased in cancer cells, compared to untransformed cells [31,32], and it is commonly accepted that malignant cells have lost the cholesterol synthesis feedback inhibition, normally induced by elevated extracellular cholesterol uptake [33]. High cholesterol levels have been associated to cancer progression [34], and increased cholesterol content of cell membranes have been associated to drug resistance, possibly due to the consequent reduced permeability [35]. For these reasons, the use of statins has been suggested as a promising therapeutic strategy for the treatment of several cancers [36], to such an extent that clinical trials have been already set up. However, a careful analysis of those trials has raised the questions of if statins have the ability to promote cancer, instead [37]. Remarkably, our data suggest that it is not the total cellular cholesterol content to affect drug resistance, but rather the balance between the endogenous and exogenous pathways. Indeed, we show that in drug-resistant HGSOC cells, cholesterol biosynthesis is reduced, while the uptake of exogenous cholesterol is increased. Accordingly, inhibition of cholesterol synthesis by statins in drug-sensitive cells induces platinum resistance, whereas reduction of lipid availability in the extracellular environment increases sensitivity in drug-resistant cells (a model schematizing these adaptations is presented in Figure 7). This phenomenon is most likely associated to the whole metabolic rewiring of this peculiar cancer type, as TRAP1 silencing alone is able to partially reproduce the transition toward a drug-resistant phenotype with reduced FDPS and OSC expression and cholesterol synthesis, and increased LDLR expression. Remarkably, the phospholipase LIPG, which may exert functions in lipid uptake from the extracellular environment and is inversely correlated to TRAP1, is also upregulated by oxidative stress and inflammation [38], which are triggered by TRAP1 downregulation [39], and we found that its silencing sensitized cell to cisplatin. Interestingly, a tight crosstalk links the Nuclear factor kappa-light-chain-enhancer of activated B cells (NF-κB) pathway and inflammatory response to Wnt/β-catenin signaling [40], a central pathway for development and differentiation playing causative role in several human diseases and in cancer. This in turn potentiates the production of cellular lipid droplets dependent on LDL-derived cholesterol and affects the endocytic pathway [41]. On the other hand, it is well known that cholesterol-rich domains termed lipid rafts and caveolae are enriched in cancer cells and dramatically increased in multidrug resistant cells [42].

As cholesterol synthesis requires oxygen, which is used for the biotransformation of squalene to cholesterol, it is conceivable that endogenous cholesterol synthesis conflicts with increased oxidative phosphorylation carried out by drug-resistant OC cells. Moreover, it has been recently shown that squalene accumulation due to loss of squalene monooxiganase expression protects anaplastic large cell lymphoma cells from ferroptotic cell death, providing survival advantage under conditions of oxidative stress [43]. Thus, auxotrophy for cholesterol is emerging as a novel strategy adopted by cancer cells to survive in specific metabolic settings. In line with the hypothesis of an increased cholesterol influx and a decreased biosynthesis, we found elevated levels of LDLR in resistant cells, and also increased expression of NPC1, a protein resident in the membrane of endosomes and lysosomes which mediates intracellular cholesterol trafficking [20]. Accordingly, NPC1 and cholesterol trafficking are also under evaluation as potential therapeutic targets in cancer [20]. Finally, alterations of lipid metabolism were shown recently in OC after anti-VEGF therapy, further underlying its potential as a target to counteract drug resistance [44]. Taken together, our data highlight the need for further research before using lipid-lowering drugs in OC, caveats for the use of statins in OC patients, and a better focus in distinguishing between endogenous production of cholesterol by cancer cells and its respective blood levels.

## Figures and Tables

**Figure 1 cells-09-00828-f001:**
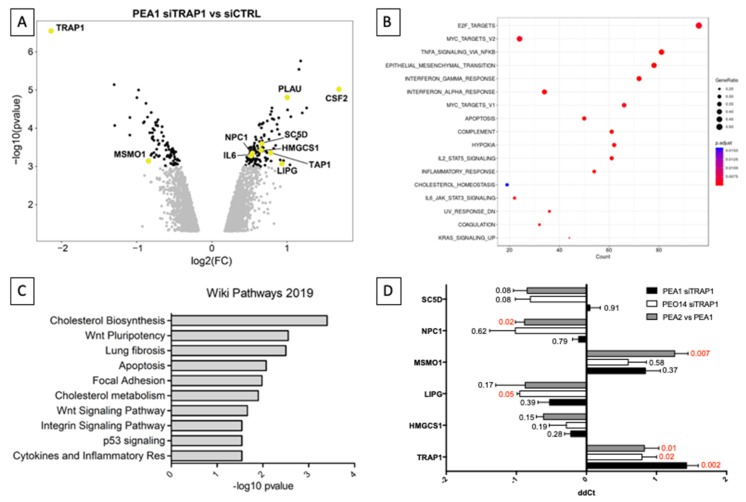
TRAP1 expression modulates genes involved in inflammation and lipid homeostasis. (**A**) Volcano plot showing the whole genome gene expression changes in PEA1 cells upon siRNA-mediated TRAP1 silencing (black = *p* < 0.001, light gray = *p*< 0.01). Relevant transcripts are highlighted, chosen for belonging to the cholesterol pathway or the inflammatory response. (**B**) Dot-plot representing hallmarks emerging from gene set enrichment analysis of the differentially expressed genes upon siRNA-mediated TRAP1 silencing in PEA1 cells. (**C**) Gene set enrichment analysis on the list of genes significantly modulated in expression (*p* < 0.001) upon TRAP1 silencing in PEA1 cells as for the microarray analysis. (**D**) Real-time RT-PCR analysis of genes involved in the cholesterol metabolic pathway found differentially expressed upon TRAP1 silencing in PEA1 cells according to microarray analyses. PEA1 and PEO14 cells were transfected with nontargeting control siRNA or TRAP1-directed siRNA (siTRAP1) and collected 72 h after transfection. All data are expressed as mean ± standard error of the mean (S.E.M.) of delta-delta Ct (ddCt) from six (PEA1 siTRAP1), three (PEO14 siTRAP1), or four (PEA2 vs. PEA1) independent experiments with technical triplicates each. Numbers indicate the statistical significance (*p*-value), based on the Student’s *t*-test (significant values highlighted in red).

**Figure 2 cells-09-00828-f002:**
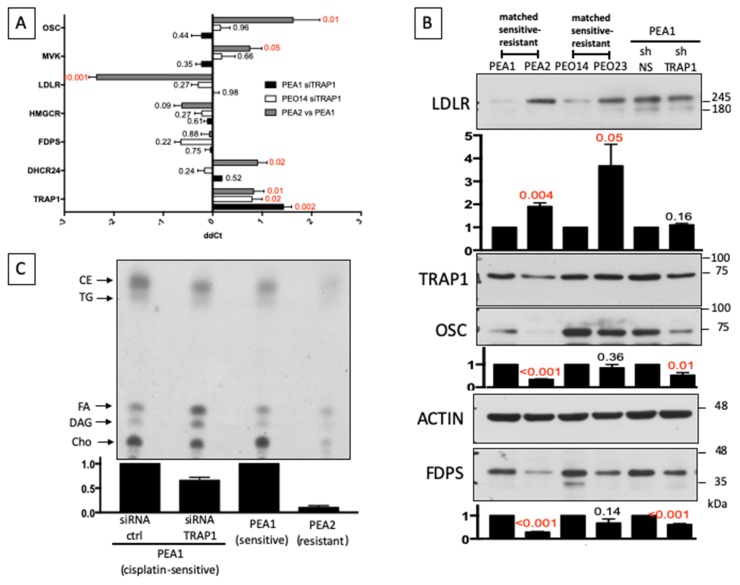
Drug-resistant cells remodel cholesterol homeostasis. (**A**) Real-time RT-PCR analysis of expression of genes belonging to the cholesterol pathway in PEA1 and PEO14 upon siRNA-mediated TRAP1 silencing and in PEA2 compared to PEA1. All data are expressed as mean ± S.E.M. of delta-delta Ct (ddCt) from six (PEA1 siTRAP1), three (PEO14 siTRAP1), or four (PEA2 vs. PEA1) independent experiments with technical triplicates each. Numbers indicate the statistical significance (*p*-value), based on Student’s *t*-test (significant values highlighted in red). (**B**) Total lysates obtained from PEA1, PEA2, PEO14, and PEO23 cells or from stable TRAP1 knockdown PEA1 cells (shTRAP1) and their respective nonsilencing controls (shNS) were separated by SDS-PAGE and immunoblotted with the indicated antibodies. Images are representative of three independent experiments. Bar plots below panels show mean ± S.E.M. of relative densitometric band intensity calculated by assuming each band of the control condition (normalized on Actin) = 1 (*n* = 3). The number above the bars represent the statistical significance (*p*-value) based on the Student’s *t*-test (significant values highlighted in red). (**C**) [^14^C]-acetate incorporation in neo-synthesized lipids. Chromatogram of non-saponifiable lipids from PEA1 cells following siRNA-mediated TRAP1 silencing and from PEA1 and PEA2 cells, radiolabeled for 2 h. Cholesterol (Cho), diacylglycerols (DAG), fatty acids (FA), triacyglycerides (TG), and cholesteryl esters (CE) are indicated with arrows. Image is representative of two independent experiments. The bar plot shows mean ± S.E.M. of relative cholesterol densitometric band intensity calculated by assuming each band of the control condition = 1.

**Figure 3 cells-09-00828-f003:**
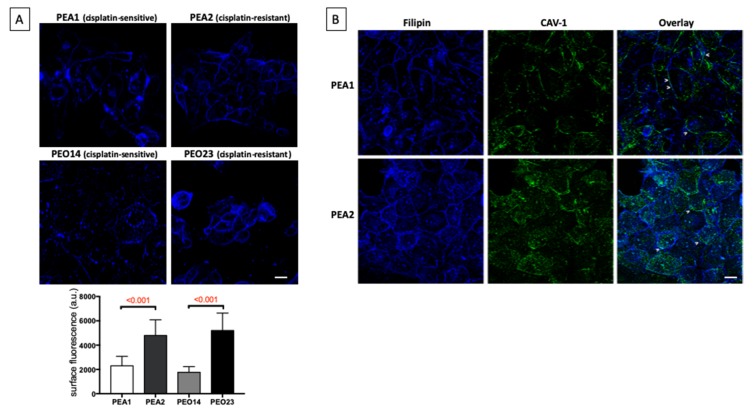
Transition to chemoresistance correlates with an increase of cholesterol at the cell surface. (**A**) Microscopy analysis of unesterified cholesterol through the visualization of filipin fluorescence in the two matched pair of cisplatin-sensitive/resistant cell lines PEA1/PEA2 and PEO14/PEO23. Maximum projection of Z-slices is shown. Scale bar = 20 μm. The bar graph (lower panel) shows the mean fluorescence intensity (arbitrary unit, a.u.) of filipin at the cell surface. Data are expressed as mean ± Standard deviation (SD) (*n* > 30). Numbers above bars indicate the statistical significance (*p*-value) based on Student’s *t*-test. (**B**) Representative images of double staining, in PEA1 and PEA2 cells, of unesterified cholesterol (Filipin) with Caveolin-1 (CAV-1). Overlay regions between filipin and CAV-1 stainings are indicated with arrowheads. Scale bar = 20 μm.

**Figure 4 cells-09-00828-f004:**
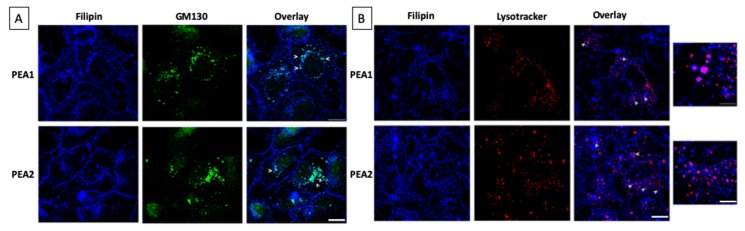
Analysis of cholesterol distribution in chemosensitive and chemoresistant cell lines. Representative images of double immunofluorescence assays. Specifically, PEA1 and PEA2 cells were stained with Filipin and the Golgi marker GM130 antibody (**A**) or the Lysostracker dye (**B**). Scale bar = 10 μm. Arrows indicate co-localization areas. In b, higher magnification pictures are shown, scale bar = 5 μm.

**Figure 5 cells-09-00828-f005:**
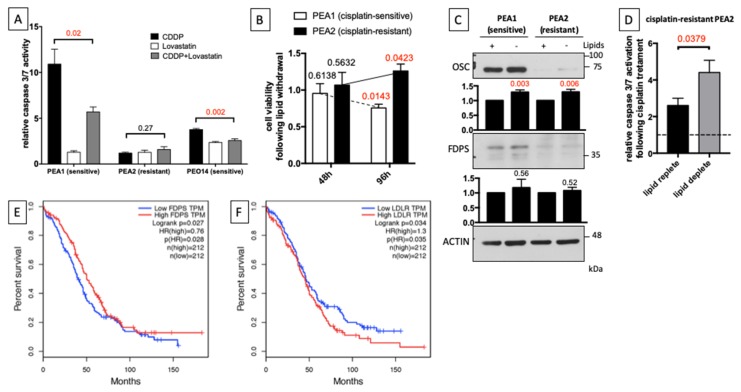
Cholesterol synthesis inhibition increases chemoresistance in platinum-sensitive OC cells. (**A**) PEA1, PEA2, and PEO14 cells were treated with 5 μM Lovastatin and then treated with 20 μM cisplatin for an additional 48 h (when medium was changed, Lovastatin was added again). Subsequently, apoptosis was measured by a luminescent caspase 3/7 activity assay. Data are expressed as mean ± S.E.M. from four (PEA1 and PEA2) or six (PEO14) independent experiments with technical triplicates each. The numbers above the bars indicate the statistical significance (*p*-value), based on the two-tailed unpaired Student’s *t*-test (significant values highlighted in red). (**B**) Twenty-four hours after seeding PEA1 and PEA2 cells in standard culture medium, cells were cultured for additional 48 or 96 h in medium supplemented with lipid-stripped serum. Finally, cell viability was measured by an MTT-based cell viability assay. (**C**) Immunoblot of total lysates from PEA1 and PEA2 cells 48 h after lipid withdrawal from culture medium. Images are representative of five independent experiments. Bar plots below panels show mean ± S.E.M. of relative densitometric band intensity calculated by assuming each band of the control condition (normalized on Actin) = 1 (*n* = 5). The numbers above the bars represent the statistical significance (*p*-value) based on the Student’s *t*-test (significant values highlighted in red). (**D**) PEA2 platinum-resistant cells were cultured for 48 h in medium supplemented with lipid-stripped serum and then treated with 40 μM cisplatin for additional 48 h. Apoptosis was measured by a luminescent caspase 3/7 activity assay. Data are expressed as mean ± S.E.M. from four independent experiments with technical triplicates each. The numbers above the bars indicate the statistical significance (*p*-value), based on the two-tailed unpaired Student’s *t*-test. (**E**,**F**) Kaplan–Meier estimates of the impact of FDPS (**E**) and LDLR (**F**) on overall survival in OC, according to The Cancer Genome Atlas (TCGA) database.

**Figure 6 cells-09-00828-f006:**
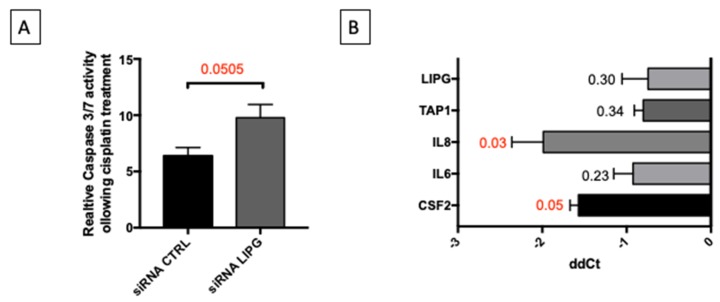
LIPG knockdown sensitizes to cisplatin and oxidative stress induces the expression of genes involved in inflammation. (**A**) PEA1 cells were transfected with control siRNAs or LIPG-directed siRNAs. Twenty-four hours after transfection, cells were treated with 20 μM cisplatin for additional 48 h. Subsequently, apoptosis was measured by a luminescent caspase 3/7 activity assay. Data are expressed as mean ± S.E.M. from four independent experiments with technical triplicates each. The numbers above the bars indicate the statistical significance (*p*-value), based on the two-tailed unpaired Student’s *t*-test. (**B**) Real-time RT-PCR analysis of expression of indicated genes in PEA1 cells treated with 20 μM H_2_O_2_ for 9 h. All data are expressed as mean ± S.E.M. of delta-delta Ct (ddCt) from four independent experiments with technical triplicates each. Numbers indicate the statistical significance (*p*-value), based on the Student’s *t*-test (significant values are highlighted in red).

**Figure 7 cells-09-00828-f007:**
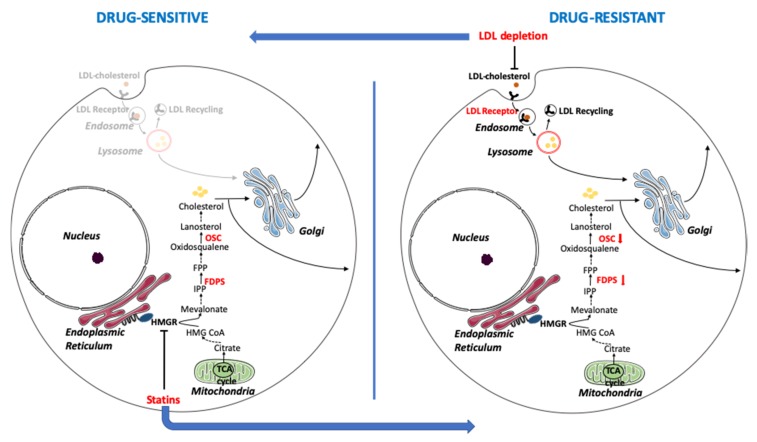
A proposed model for the remodeling of cholesterol metabolism in the transition from a chemosensitive to a chemoresistant phenotype in human ovarian cancer cells. Drug-sensitive cells show significant activation of the cholesterol biosynthetic pathway and transport of cholesterol through the Golgi apparatus, while uptake of exogenous cholesterol through the LDLR is limited. Drug resistant cells, on the opposite, show reduced levels of FDPS and OSC along the biosynthetic pathway and increased LDL uptake. The inhibition of 3-hydroxy-3-methylglutaryl-coenzyme A reductase (HMGCR), the rate-limiting enzyme of the biosynthetic pathway, by statins induces drug resistance, whereas the reduction of LDL recycling and lipid absorption through extracellular lipid withdrawal or LIPG knockdown sensitizes cells to drug-induced apoptosis. Enzymes whose expression has been analyzed in this work are in bold red. IPP = isopentenyl pyrophosphate; FPP = farnesyl pyrophosphate; TCA = tricarboxylic acid. This image was created using images from Servier Medical Art under Creative Commons Attribution 3.0 Unported License (https://smart.servier.com).

**Table 1 cells-09-00828-t001:** Primers used for quantitative PCR analyses. CSF2: Colony Stimulating Factor 2; DHCR24: 24-dehydrocholesterol reductase; FDPS: Farnesyl Diphosphate Synthase; HMGCR: 3-hydroxy-3-methylglutaryl-coenzyme A reductase; HMGCS1: 3-Hydroxy-3-Methylglutaryl-CoA Synthase 1; IL6: Interleukin 6; IL8: Interleukin 8; LDLR: Low Density Lipoprotein Receptor; LIPG: Lipase G; MVK: Mevalonate kinase; MSMO1: Methylsterol Monooxygenase 1; NPC1: Niemann-Pick C1 Protein; OSC: oxidosqualene cyclase; SC5D: Sterol-C5-Desaturase; TAP1: Transporter associated with Antigen Processing 1; TRAP1: Tumor necrosis factor Receptor Associated Protein 1.

Primer ID	Forward (5′-3′)	Reverse (5′-3′)
ACTIN	CCTCACCCTGAAGTACCCCA	TCGTCCCAGTTGGTGACGAT
CSF2	GTCATCTTGGAGGGACCAAGG	CCATGCCTGTATCAGGGTCAG
DHCR24	TGAAGACAAACCGAGAGGGC	CAGCCAAAGAGGTAGCGGAA
FDPS	AGCAGGATTTCGTTCAGCAC	TCCCGGAATGCTACTACCAC
HMGCR	TACCATGTCAGGGGTACGTC	CCAGTCCTAATGAAACCTTAGAAG
HMGCS1	TGTCCTTTCGTGGCTCACTC	GCCAGCAAGCTTCTGCATTC
IL6	TGCAATAACCACCCCTGACC	CAATCTGAGGTGCCCATGCT
IL8	CCAGTCTTGTCATTGCCAGC	TGACTGTGGAGTTTTGGCTGT
LDLR	GAATTTGGCCAGACACGGT	CACCGTACCCAGCTGATTTT
LIPG	CAGGCTGTGGACTCAACGAT	TCGGCTTGTCCTGATTCACC
MVK	GCTCAAGTTCCCAGAGATCG	ATGGTGCTGGTTCATGTCAA
MSMO1	GGTTCCGAGGTTGGAACACCT	TTCAAATCTCTGCAGACAGCCT
NPC1	TTGTGGTGTTGGCTTTTGCC	GTTCGCGCTCTGTTCCTTTG
OSC	ATGGTGGCCCACTTTTCCTC	CTGCACTGACCGCAGGTA
SC5D	TTTGCAGAGCAGTGGCGT	CTGGCCATGTGGCTGGATAC
TAP1	CCTCCTTTCCAAGCTCCTCG	GGCCAGCATATGCCTTCAGT
TRAP1	GACGCACCGCTCAACAT	CACATCAAACATGGACGGTTT

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
