# Peer review of "Cholesterol Homeostasis Modulates Platinum Sensitivity in Human Ovarian Cancer"

_cells, 2020, doi:10.3390/cells9040828_

Round 1
Reviewer 1 Report
Criscuolo et al. describe the effects of cholesterol homeostasis modulation and sensitivity of human ovarian cancer to platinum. This point is important as cisplatin treatment is often associated to multiple resistance mechanisms. The evidence suggested by the authors is that cholesterol metabolism could one of the key mechanism of the drug resistance. The understanding of this mechanism is thus of a great importance for clinical improvement. Even though interesting, the manuscript lacks some control and/or replicates, which sometimes induces an over interpretation of the results in the manuscript. The following comments are thus provided to improve the quality of the manuscript.
Mayor comments:
1) One important point is that the authors have uses the one-sample t test, a paired test variant. However, the comparison of a cell line in two different conditions (transfected vs. scramble) as well as the comparison of different cell lines (PEA1 vs. PEA2 or PEA1 vs. PEO24) should be treated with an unpaired-test. Authors should thus analyze all their data with a new statistic test and check whether significance has been changed.
2) Numerous discrepancies between the comments in the main text and the results shown in the figures have been found. Following are some examples:
2.1. For PEA1-TRAP1siRNA the authors claim that regulation of various genes was validated by qPCR (line 206, fig 1D). However, RNAseq data from PEA1-TRAP1siRNA show an increase in the expression of SCD5, HMGSC1, NPC1 that are not observed in qPCR analysis. Please comment or amend the manuscript.
2.2. Fig 2, and as previously, qPCR analyses of PEA1-TRAP1siRNA does not show any change in the messenger levels of FDPS, LDLR and OSC, while protein levels seem modified. This is not surprising as proteins and their respective mRNA do not always similarly accumulate. The authors should comment that point. Besides, western blot should be done at least in triplicate to allow a real quantification of the protein accumulation, and finally to conclude regarding the variations suggested in Fig. 2.
2.3 Line 334-335: “lipid deprivation increased caspase 3/7 activation following cisplatin treatment (40 μM) (Fig.4D)”. Figure indicates a p-value 0.1418, even though it could (eventually) be considered as a tendency, it cannot ascertain the written sentence, which should thus be modulated in its conclusion. Line 363-365: the same comment is done “LIPG silencing by two different siRNA sequences increased caspase 3/7 activity upon cisplatin treatment (Fig.5A), despite the fact that figure indicates p-values of 0.1173 and 0.2079 in the analysis vs siRNA control, respectively. There is thus no significant change and the comment should be changed. Line 370 “Results show that, indeed, all of the genes mentioned above are significantly induced upon treatment”. However, the figure shows p ≥ 0.05 for IL6, TAP1 and LIP-G. Again no significant change is pointed out.
Altogether, these points raise the question of what was the P-value considered as significant? Are there real changes or trends? Do trends support the conclusion? Authors should clarify these points.
3) The comment for Fig. 3C (line 292-296) indicates that ”chemo resistant cells and TRAP1-knockdown cells show clear accumulation of cholesterol on the cell membrane”. Based on the images, it is quite challenging to have the same opinion as the authors: no quantification has been performed, which is not surprising as immunofluorescence cannot be quantified; the various cell compartments are difficult to identify. The authors should at least co-stained membranes proteins or endosomal proteins type Rab5, Cav1 or Flot2 with specific antibodies.
4) Results shown in Figure 3A should be performed in triplicate to allow a quantification to get robust data.
5) Caspase 3/7 activity assays should be easily correlated/validated with a MTT or a trypan blue assay.
6) The cell accumulation of cholesterol is supposed to be highly toxic. One could thus wait for the activation of mechanisms that regulate this accumulation. The quantification of the esterified cholesterol and the accumulation of lipid droplets would give a sort of confirmation of this accumulation. Another indirect way to confirm this accumulation would be to measure the levels of SREPBP2, ABCs… proteins or their corresponding mRNA.
Minor comments:
* Throughout the discussion, the Wnt pathway seems to have an important role. It could be suggested to discuss the possible role of this pathway in the downregulation of the cholesterol biosynthesis.
* Likewise, the modifications of the cholesterol homeostasis and the modification of the signaling pathways associated to the lipid rafts could be discussed in the context of the drug resistance?
* An interesting point would be to understand what could happen in paclitaxel resistance or in combined cisplatin/taxol treatment. PARP inhibitors have been shown to have good clinical outcomes. Would it be interesting to analyze whether the drug resistance in PE2, PEA1-siTRAP1 or PE014 cells is maintained with PARP inhibitors?
* English language should be reviewed by a native speaker.
Reviewer 2 Report
The authors describe a novel mechanism by which cancer cells acquire chemotherapy resistance through the downregulation of cholesterol biosynthesis. The experiments are novel and well controlled. Presentation can be improved to make the results clearer to the reader.
Major comments:
Much of the confusion stems from the sheer number of targets that are mentioned in the study. The central targets should be included in the summary figure, which serves as a good reference. It is difficult to keep track of the cell line models. Please indicate in the figures which cell lines are sensitive/resistant and which cell lines are paired. Figure 3A, results are not clear. Would be helpful to show the entire picture with the additional bands for other forms of cholesterol. Also, showing the positive control would be helpful. Figure 3B, images do not show up well when printed. Pictures should be replaced with clearer images. In the text localization of the cholesterol is discussed and this is not evident in the pictures. What is the effect of lovastatin and lipid withdrawal on proliferation? Cells with lower rates of proliferation are inherently more resistant to cisplatin. In figure 5 the sensitive PEA1 cell line is used. Are similar effects seen in a resistant cell line?Author Response
Please see the attachment

Round 2
Reviewer 1 Report
The authors have adequately responded to the various comments and the quality of the manuscript has been improved.
Author Response
We thank the reviewer for the positive assessment.
We have edited the manuscript to amend minor style/language errors.